# Investigating Rural Domestic Waste Sorting Intentions Based on an Integrative Framework of Planned Behavior Theory and Normative Activation Models: Evidence from Guanzhong Basin, China

**DOI:** 10.3390/ijerph17134887

**Published:** 2020-07-07

**Authors:** Jing Shen, Donghui Zheng, Xiaoning Zhang, Mei Qu

**Affiliations:** College of Economics and Management, Northwest A&F University, 3 Taicheng Road, Yangling 712100, China; arya_shen@163.com (J.S.); zdh@nwafu.edu.cn (D.Z.); xiaoningzhang@nwafu.edu.cn (X.Z.)

**Keywords:** planned behavior theory, normative activation theory, farmer, domestic waste sorting, Guanzhong Basin, China

## Abstract

The sorting of domestic waste is the most effective way to alleviate the problem of mass garbage accumulation around the villages of rural China. Farmers are the creators of rural domestic waste as well as the direct beneficiaries of effective waste management. However, few studies have been conducted on the psychological determinants of farmers’ intentions to sort domestic waste. This paper applies planned behavior theory (TPB) and normative activation theory (NAM), to analyze the domestic waste sorting intentions of rural residents in Guanzhong, China. Based on the micro-data of 327 rural households in Guanzhong, structural equation models of the factors influencing farmers’ domestic waste sorting intentions were estimated. The results demonstrate the following. (1) Farmers’ attitudes, perceived behavioral control, and personal norms have significant positive direct impacts on their domestic waste sorting intentions, with personal norms having the greatest direct impact. (2) Subjective norms have no direct impact on farmers’ domestic waste sorting intentions but do have an indirect impact on them through personal norms, behavioral attitudes, and perceived behavior control. This article increases scholarly understanding of the psychosocial determinants of the environmentally friendly behavioral intention to sort domestic waste. The study also provides academic and theoretical support to policy makers in implementing relevant policy recommendations.

## 1. Introduction

Environmental pollution caused by the accumulation of domestic waste is a serious problem in rural areas all over the world [1]. Especially in developing countries, due to the continuous development of rural economies and changes in rural residents’ lifestyles, the production of rural domestic waste has increased significantly [2]. Due to the lack of waste management facilities and appropriate waste management methods in rural areas, poor management of domestic waste leads to the destruction of rural ecological environments and increasing negative impacts on the life and health of rural residents [3,4,5]. Rural domestic waste management in China, one of the world’s major developing countries, has become a major challenge for the Chinese government. At present, nearly one-quarter of rural domestic waste in China is not being collected and treated [6]. The current method of domestic waste disposal promoted in rural China is “household collection, village concentration, town transfer, and county (city) treatment [6].” However, most rural domestic waste collection methods are mixed, without prior classification [7]. Whether domestic waste is disposed of at a waste incineration power plant or a waste terminal treatment landfill, it is difficult to use the resources fully and to reduce disposal. Effectively dealing with rural domestic waste is one of the most important problems in worldwide rural development.

Sorting at source is the most effective way to ease the rural domestic waste problem [8,9]. Doing so can not only reduce the cost of end-of-life disposal of domestic waste but also improve resource utilization efficiency. A series of policies and regulations concerning the disposal of rural domestic waste has been issued by the Chinese government. For example, in November 2015 the Chinese government proposed, in the “Guiding Opinions on Comprehensively Promoting Rural Waste Management,” that China’s rural areas should comprehensively manage domestic waste and promote the reduction of domestic waste at source. In December 2016, the General Office of the State Council of the People’s Republic of China issued an article entitled, “Several Opinions on Deepening the Advancement of Structural Reforms on the Agricultural Supply Side and Accelerating the Development of New Momentum for Agricultural and Rural Development,” which mentioned that China should vigorously promote special action for rural domestic waste treatment along with waste sorting and resource utilization. In February 2018, “Rural Waste Management” was listed as one of the six key tasks in the “Three-year Action Plan for Rural Habitat Environment Improvement” issued by the Central Office of the Communist Party of China and the General Office of the State Council of China. This series of policies reflects the importance of effective management of rural domestic waste in improving the ecological environment in rural China and emphasizes its urgency.

Sorting rural domestic waste is an effective measure for improving the long-term operation mechanism of rural domestic waste control. The effective treatment of rural domestic garbage requires the active participation of farmers. Therefore, it is important to analyze farmers’ intentions to sort waste and the factors that affect their waste sorting, in order to allow the implementation of appropriate intervention strategies in the future to enhance farmers’ environmentally friendly behavior. A review of existing literature reveals that many studies have assessed the factors that influence garbage sorting behavior; however, most focus on urban garbage sorting, whereas little research focuses on the sorting and the determinants of domestic garbage in rural areas. In addition, the intention to sort and recycle waste is affected by various factors apart from external factors, such as government propaganda and education [10,11], rewards and punishments [12], systems and regulations [13], and hardware conditions of service facilities [14]. Many scholars have proposed that social demographic variables such as gender, age, and education level, also affect waste sorting and recycling behavior [15,16,17]. However, studies of individual intentions to engage in behavior that protects the environment find that external contextual factors and socio-demographic variables can only explain a small part of the variance [18], while socio-psychological variables are important predictors of participation in pro-environmental behaviors [19,20]. Therefore, the purpose of this study is to focus on the impact of socio-psychological factors on farmers’ intentions to sort domestic waste and to provide a theoretical basis for the formulation of rural domestic waste sorting management policy by the Chinese government.

## 2. Literature Review and Research Hypothesis

### 2.1. The TPB Framework

The planned behavior theory (TPB) is a classic theoretical framework that academics use to study various environmental behaviors [21,22], such as choice of travel modes [23], organic food purchases [24], use of bio-energy [25], green hotel consumption behavior [26], household waste recycling [27,28], and citizen environmental complaints [29], among others. The TPB provides a useful and reliable theoretical framework that can be used for the systematic exploration of factors that influence individual waste sorting decisions [30]. Ajzen [31] proposed that the only direct psychological determinant of human behavior is intention, whereas attitudes, subjective norms, and perceived behavior control are three independent factors that can predict the occurrence of intentions. Here, attitude refers to “positive or negative evaluations produced by individuals when they perform behaviors”. Subjective norms are “social pressures from family members, neighbors, and others that individuals perceive when performing certain behaviors”. Finally, perceived behavior control is “individual’s understanding of their behavior“. 

In general, more positive attitudes and subjective norms, in conjunction with perceived behavioral control, can enhance an individual’s intention to perform a specific behavior [32,33]. According to the assumptions underlying the TPB, if farmers have a positive attitude towards the sorting of domestic waste, feel the support of family or neighbors for garbage sorting, and think that they have the time and energy to engage in waste sorting, then they will be more willing to participate in domestic waste sorting. In addition, research shows that subjective norms have an indirect effect on behavioral intentions through behavioral attitudes and perceived behavioral control [29,34]. Specifically, the attitude of farmers toward performing a domestic waste sorting behavior will be affected by the views of family members or neighbors, and the thoughts of family members or neighbors will affect the farmers’ perception of whether the sorting behavior is easy to implement, thus affecting farmers’ intention to engage in domestic waste sorting. Therefore, the following five hypotheses are proposed:

**Hypothesis 1 (H1)**.
*A farmer’s behavioral attitude positively affects their domestic waste sorting intention.*


**Hypothesis 2 (H2)**.
*A farmer’s subjective norms positively affect their domestic waste sorting intention.*


**Hypothesis 3 (H3)**.
*A farmer’s perceived behavioral control positively affects their domestic waste sorting intention.*


**Hypothesis 4 (H4)**.
*A farmer’s subjective norms positively affect their attitude towards domestic waste sorting.*


**Hypothesis 5 (H5)**.
*A farmer’s subjective norms positively affect their perceived behavioral control of domestic waste sorting.*


### 2.2. The NAM Framework

In 1977, Schwartz proposed the normative activation model (NAM), which can effectively predict public environmental behavior. This theoretical model has been used to predict various prosocial and environmentally friendly behaviors, such as energy use [35], citizens’ environmental complaint behavior [29], household waste sorting [36], public transportation use [37], and individual energy conservation [38]. The NAM includes three core variables: consequence awareness, attribution of responsibility, and personal norms [39]. Consequence awareness consists of “the cognition that an individual believes that failure to perform a specific behavior may bring adverse consequences to others” [39]. Attribution of responsibility consists of “the individual’s sense of responsibility for the consequences of not performing a particular behavior” [39]. Personal norms are “individuals’ perception of their own moral obligations to perform a certain behavior” [39]. Personal norms are important factors influencing behavioral intentions [29,37].

According to the NAM, when individuals carry out certain environmental behaviors that may adversely affect society or others, they will think that they have caused negative consequences, and they will perceive their own responsibility and form their own moral obligations. Thereafter, they would be more willing to implement environmentally friendly behaviors [29,40]. The sorting of domestic waste is a pro-environmental behavior that could promote resource recovery and improve resource utilization [41]. If the garbage is directly processed without sorting, it would also cause many adverse effects, such as waste of resources, soil pollution, and disease [6]. If farmers are more aware of the negative consequences caused by the direct treatment of domestic waste, they are more willing to take responsibility for facing these negative consequences. They would be more convinced of their moral obligation to sort garbage, thereby forming a positive personal norm that increases their domestic waste sorting intention. Therefore, the following hypotheses are proposed:

**Hypothesis 6 (H6)**.
*A farmer’s awareness of the consequences positively affects their ascription of responsibility for domestic waste sorting.*


**Hypothesis 7 (H7)**.
*A farmer’s ascription of responsibility positively affects their personal norms regarding domestic waste sorting.*


**Hypothesis 8 (H8)**.
*A farmer’s personal norms positively affect their intention toward domestic waste sorting.*


### 2.3. The Integrated Framework of TPB & NAM

The extensive application of TPB and NAM in the field of environmental protection has provided a deep understanding of the environmental behavior of individuals. Existing research has combined the two theories as the key framework for explaining environmental behavior. It also has shown that models that integrate TPB and NAM can better predict and explain individual environmental behavior than either theory can predict by itself [29,34]. The difference is that TPB posits that environmental behavior emerges from the individual’s own expectations and welfare, whereas NAM pays more attention to the individual’s altruistic behavior and moral norms. The internal factors of the two theories affect each other and, ultimately, the individual’s intention to engage in environmental behavior [29].

Specifically, farmers’ recognition of adverse consequences will be conducive to a positive attitude and a more pro-environmental subjective norm [29,34]. Thus, if farmers think that sorting domestic waste can reduce pollution and reduce the use of resources, they will have a more positive attitude toward implementing this behavior. When farmers know more about the impact of garbage sorting on humans and the environment, they will better understand others’ expectations and the resulting social pressures on the behavior of garbage sorting, thereby increasing more favorable subjective norms. While a subjective norm is a pre-variable of a personal norm [42] and it has positive effect on acting on the ascription of responsibility [29], a subjective norm can also verify the correctness of an individual’s environmental behavior and guide individuals to ascertain whether their beliefs are beneficial to themselves [42]. Increasing pressure or expectations on household members to sort domestic garbage makes it easier to form a sense of responsibility for domestic garbage sorting and, therefore, creates stronger personal norms supporting this behavior. Therefore, the following hypotheses are proposed:

**Hypothesis 9 (H9)**.
*A farmer’s awareness of consequences positively affects their attitude towards domestic waste sorting.*


**Hypothesis 10 (H10)**.
*A farmer’s awareness of consequences positively affects their subjective norm with respect to domestic waste sorting.*


**Hypothesis 11 (H11)**.
*A farmer’s subjective norms positively affect their ascription of responsibility for domestic waste sorting.*


**Hypothesis 12 (H12)**.
*A farmer’s subjective norms positively affect their personal norm with respect to domestic waste sorting.*


Based on the above theoretical analysis, the research framework proposed in this paper is shown in Figure 1.

## 3. Methodology

### 3.1. Measures and Questionnaire Design

In this paper, farmers’ attitudes, subjective norms, and personal norms related to domestic waste sorting are latent variables, which cannot be observed or measured directly and are instead indirectly measured through other specific observable indicators [43]. Based on TPB and NAM, the questionnaire drew on indicators developed by other scholars [28,44] and included the actual sorting of rural domestic waste in China to determine the draft measurement items. Afterward, the researchers discussed the questionnaire with experts and scholars in ecological environment governance to delete or replace inappropriate items. A preliminary survey was conducted in the countryside around the school, and some items were corrected or deleted accordingly in order to determine the final specific measurement items for each variable. All items were measured on a five-point Likert scale where 1 = “strongly disagree,” 2 = “disagree,” 3 = “neutral,” 4 = “agree,” and 5 = “strongly agree.” See Table 1 for details.

### 3.2. Data Collection and Samples

The data collected for this paper came from a household survey of rural areas in Heyang County in China’s Guanzhong Plain; the survey was administered in July 2019. Heyang County is located at 109°58′~110°25′ E, 34°59′~35°26′ N (Figure 2) and serves as a model county for beautiful rural construction in China, rural revitalization, and improving rural settlements in Shaanxi Province. The distance from the north end to the south end of Heyang County is 41.8 km; the distance from the east end to the west end is 35.6 km; and the county has a total area of 1437 km^2^ and a total population of 510,000. Since 2018, Heyang County has actively responded to national policies committed to improving the ecological environment of rural settlements and implemented effective treatment of rural domestic waste. For this study, a survey was conducted in the three townships of Heyang County: Tongjiazhuang in the north, Fangzhen in the center, and Lujing in the south. The investigation found that that all households bring their domestic waste to a designated place for centralized processing. Among them, 25.08% of households reported not sorting their waste at all, while 74.92% only sorted waste with economic value (mainly newspapers, old books, cans, plastic bottles, scrap metal, etc.).

Before the formal survey, the authors conducted a pre-survey of a sample of 30 subjects in rural areas around the school to ensure that the questionnaire was valid and usable. Thereafter, the investigators went to the farmers’ homes in Heyang County to conduct face-to-face interviews. As a result, a total of 350 questionnaires were completed. After sorting and deleting the questionnaires containing incomplete entries, 327 questionnaires were finally used for data analysis. The questionnaire’s effective rate was 93.43%. In the effective sample of this survey, most respondents were female (58.7%); 67.5% of the respondents were aged 46–65. More than half of the respondents (58.7%) had between four and six family members. The vast majority of the respondents (87.8%) had a low level of education, that is, junior high school education and below. The annual household income of the respondents was below RMB 80,000 (11,276 US dollars) for 69.7% of the total sample; 79.2% of the farmers in the sample area were mainly engaged in agricultural production activities. 

### 3.3. Analytical Method

Structural equation modelling (SEM) was used to test the hypothetical relationships between the variables in the proposed TPB and NAM integrated model. In this study, data were analyzed by maximum likelihood estimation using SPSS 23.0 (IBM, Armonk, New York, NY, USA) and AMOS 22.0 (IBM, Armonk, New York, NY, USA). Confirmatory factor analysis was used to estimate the measurement model to ensure that the measurement factors and scale items were highly correlated with each other [33]. Cronbach’s alpha (α) was calculated to evaluate the internal consistency between the measurement items for each latent variable. Convergence validity was evaluated according to the factor load of each observation index; when testing reliability, the combined reliability (CR) should be greater than 0.6 [42] and the average variance extraction (AVE) should be greater than 0.5 [45]. Next, the structural model was used to test the hypothetical relationships between the variables and evaluate the overall goodness of fit of the model. The goodness of fit indicators generally selected are chi-square (c^2^), adjusted goodness fit index (AGFI), comparative fit index (CFI), goodness fit index (GFI), parsimony normed fit index (PNFI), Tucker-Lewis index (TLI), normed fit index (NFI), and robustness of mean squared error approximation (RMSEA). The PNFI index should be greater than 0.5. The NFI, CFI, GFI, TLI and AGFI should exceed 0.9, and the RMSEA ideally be greater than 0.05 and lower than 0.08 [46]. Finally, in order to examine further the degree of interaction between the latent variables in the integrated model of TPB and NAM, a Bootstrap test was performed on the model using AMOS software. 

## 4. Results and Discussion

### 4.1. Measurement Model Analysis

The test results are shown in in Table 2. First, the Cronbach’s α for each latent variable was between 0.675 and 0.851, and the CR values were between 0.6862 and 0.8481; all of these values exceeded 0.6, indicating good internal consistency and reliability between the items for each latent variable. Second, the normalized factor loadings of the 14 observed variables were between 0.634 and 0.910, indicating that each observed variable had a sufficient level of explanatory power for its latent variable. The AVE values of all of the latent variables exceeded 0.5, indicating that the convergence validity and reliability of the measurement model were good. In addition, Table 3 shows that the square root of the AVE value of each latent variable was greater than the correlation coefficient between other latent variables, demonstrating discriminant validity [43].

### 4.2. Structural Model Analysis

The fitness indices of the measurement model were CMIN/DF = 2.013, PNFI = 0.677, NFI = 0.947, CFI = 0.972, TLI = 0.961, GFI = 0.945, AGFI = 0.911, and RMSEA = 0.056, all of which are in accordance with accepted standards, indicating that the measurement model fit the data well.

The SEM results are shown in Table 4 and illustrated in Figure 3. First, within the framework of TPB, behavioral attitude and perceived behavior control positively affected domestic waste sorting intention at the 1% and 5% significance levels. Hypotheses 1 and 3 were therefore supported. This shows that positive attitudes and the perception of behavior as easy to implement can promote the intentions of farmers to participate in domestic waste sorting, which is consistent with Xu et al. [15], Davis et al. [27], and Kang et al. [47]. Since the standardized path coefficient of subjective norms for behavioral intention, −0.055, was not significant, Hypothesis 2 was not supported. This is consistent with the findings of Bamberg and Mser [37]. When variable personal norms were added to the model, subjective norms no longer had a direct impact on farmers’ domestic waste sorting intentions. In the results, the standardized coefficients of subjective norms on behavioral attitude and perceived behavior control were 0.407 (*p* < 0.001) and 0.773 (*p* < 0.001), respectively, which indicated that subjective norms positively and significantly influenced behavioral attitude and perceived behavior control; thus, Hypotheses 4 and 5 were supported. Although subjective norms do not directly affect domestic waste sorting behavioral intentions, they can indirectly affect farmers’ domestic waste sorting intention through attitude, perceived behavioral control, and personal norms. Subjective norms strengthen farmers’ domestic waste sorting intentions by stimulating their positive attitude and strong personal moral obligations. Moreover, they play a key role in farmers’ domestic waste sorting, as confirmed in research by other scholars [34,37]. Second, in the framework of NAM, the coefficients for H6 (β = 0.425, *p* < 0.001), H7 (β = 0.323, *p* < 0.001), and H8 (β = 0.594, *p* < 0.001) were each statistically significant; therefore, Hypotheses 6, 7, and 8 were supported, indicating that a stronger awareness of consequences among farmers will promote a higher sense of moral responsibility and, therefore, activate their personal norms and influence their intentions and behavior. The findings of Lopes et al. [48] and Nguyen et al. [49] also confirm this. Third, in the integrated TPB and NAM model, the internal factors of the two theoretical models interacted with each other. On the one hand, awareness of the consequences in NAM significantly positively affected behavioral attitudes and subjective norms in TPB, supporting Hypotheses 9 and 10. On the other hand, the subjective norm of domestic waste sorting by farmers in TPB positively affected the awareness of the consequences and the personal norms in NAM; thus, Hypotheses 11 and 12 were supported.

### 4.3. Mediation Effects Analysis

The Bootstrap test in AMOS can reflect the direct, indirect, and overall impact of each variable on sorting intent in the integrated TPB and NAM model, thereby further verifying the important influence of personal norms, subjective norms, and other factors on the farmers’ intentions toward domestic waste sorting. As shown in Table 5, personal norms were the strongest direct determinants of farmers’ domestic waste sorting intentions, which is consistent with the results of Bortoleto et al. [50] and Guo et al. [44]. When farmers choose whether to sort domestic waste, personal norms play a key role. A greater awareness among farmers of their responsibilities and obligations in implementing waste sorting was associated with a greater tendency to sort domestic waste. Although subjective norms had no significant direct impact on farmers’ domestic waste sorting intentions, they indirectly promoted their intention through attitudes, perceived behavioral control, and personal norms, providing further support for the SEM results. This also indirectly highlights the importance of subjective norms. In the integrated TPB and NAM model, subjective norms no longer directly affected farmers’ domestic waste sorting intentions; instead, both farmers’ positive attitudes toward waste sorting and strong personal moral obligations enhanced their intention toward domestic waste sorting.

The research presented in this article has several implications for encouraging farmers to sort domestic waste in their daily lives. As personal norms had the greatest direct impact domestic waste sorting intentions, incentive policies such as “honorary farmers” can be adopted to encourage farmers to participate actively in domestic waste sorting, to emphasize that domestic waste sorting is a correct and worthwhile environmental behavior, and to enhance farmers’ awareness of the necessity and importance of sorting household waste, so as to enhance their sense of moral obligation and motivate them to act accordingly [7,50,51]. At the same time, subjective norms had the greatest overall impact on farmers’ intentions to sort household waste. Therefore, in addition to making full use of traditional and modern media, the government should also encourage celebrities in villages to take the lead in encouraging their families and relatives to sort domestic waste [28,36,49]. This can set an example and create a good social environment for environmental protection, thereby increasing social pressure on farmers and promoting farmers’ intention to sort domestic waste. 

## 5. Limitations and Way Forward

This paper applies the comprehensive theoretical framework of TPB and NAM in examining the factors that influence domestic waste sorting behavioral intentions. The results verify that farmers’ attitudes toward domestic waste sorting, subjective norms, perceived behavioral control, and personal norms all affect their intention to sort domestic waste. The comprehensive theoretical framework combines various factors from the two theories and strengthens the understanding of the factors affecting farmers’ domestic waste sorting behavioral intentions. The comprehensive TPB and NAM model explains 70% of the variance in farmers’ domestic waste sorting intention.

There are some deficiencies in this study. First, this paper has focused on farmers’ behavioral intentions regarding domestic waste sorting. Although intention is a good predictor of behavior, it cannot fully represent actual behavior. Therefore, in the future, the actual behavior of farmers’ domestic waste sorting should be further studied. Second, there are many factors, both internal and external, that affect farmers’ intentions regarding domestic waste sorting. In addition to the factors mentioned in this study, other influencing factors, such as environmental values [11], perceived policy effectiveness [52], penalty mechanisms [7], and facility convenience [28], may affect farmers’ behavioral intention with regard to domestic waste sorting. Further research can add these factors to the conceptual framework and conduct empirical tests to improve the comprehensive study of influencing factors. Finally, the sample size of this study is limited. Considering that the actual situation of domestic waste sorting in different types of rural areas may be different, more sample data can be collected from a broader research scope in future research to analyze the domestic waste sorting problems of rural residents in different areas.

## 6. Conclusions

Rural domestic waste is a major problem for the ecological environment of rural human settlements in China, thereby requiring urgent solutions. Sorting domestic waste at its source is an effective method to solve this problem. However, few studies have analyzed the social and psychological factors affecting farmers’ waste sorting behavior. Based on the integrated framework of TPB and NAM, this article has explained the psychological factors affecting the intention to engage in domestic waste sorting behavior. In general, the integrated TPB-NAM framework has received sufficient support. The integrated TPB and NAM model explained 70% of the variance in farmers’ intentions to sort domestic waste. The main conclusions of this paper are as follows. First, attitude, perceived behavior control, and personal norms all directly and positively affect farmers’ domestic waste sorting intentions; of these, personal norms have the greatest direct impact. Second, subjective norms have no direct impact on farmers’ domestic waste sorting intention but have an indirect impact on them through personal norms, behavioral attitudes, and perceived behavior control.

Based on these conclusions, the following three policy recommendations are proposed. First, the government should pay attention to cultivating farmers’ environmental ethics; it should strengthen farmers’ responsibility to protect the environment and conserve resources and publicize information on domestic waste sorting through various channels, such as the TV and the Internet. The government should also improve farmers’ acceptance of domestic waste sorting policy and attitudes toward active participation, deepen farmers’ awareness of and responsibility attribution for domestic waste sorting results, promote the formation of personal norms, and develop domestic waste sorting habits. Second, the government should improve the perceived behavior control of farmers, decrease the perceived difficulties of domestic waste sorting, and reduce the obstacles to waste sorting that they expect to face. For example, the construction of supporting facilities for rural domestic waste sorting could be improved and the quality of domestic waste sorting services could be promoted. Third, making use of relationship networks, village committees and village celebrities should lead by example to mobilize their neighbors to participate actively in the sorting of domestic waste. Doing so would create the social pressure on everyone to sort waste and would thereby strengthen farmers’ subjective norms in order to increase their domestic waste sorting intention.

## Figures and Tables

**Figure 1 ijerph-17-04887-f001:**
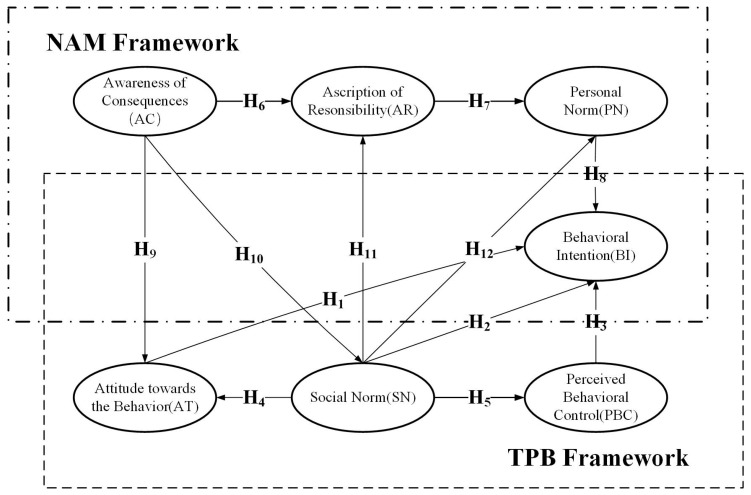
Theoretical analysis framework.

**Figure 2 ijerph-17-04887-f002:**
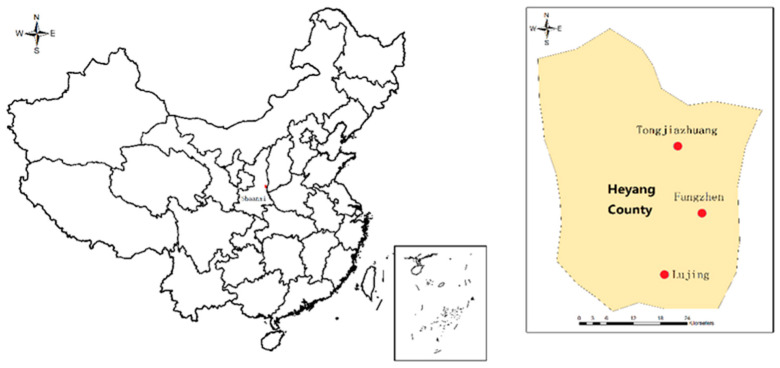
Regional distribution.

**Figure 3 ijerph-17-04887-f003:**
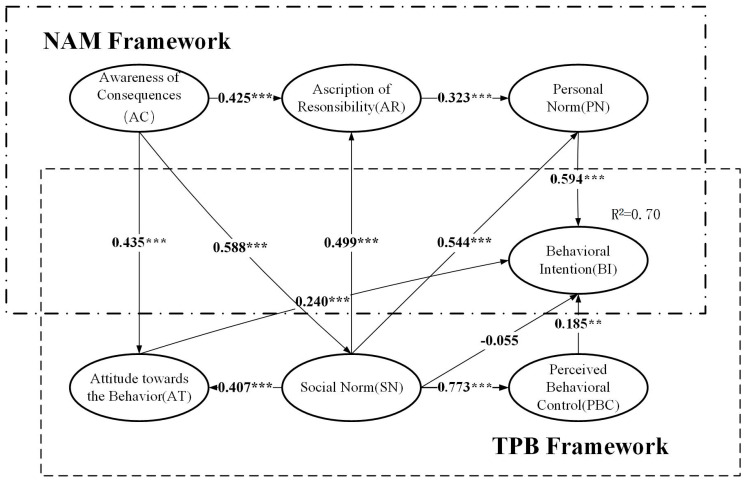
Results of model estimation. *** *p* < 0.01, ** *p* < 0.05.

**Table 1 ijerph-17-04887-t001:** Measurement items on domestic waste sorting attitude and behavior

Constructs	Measurement Items
Awareness of Consequences (AC)	Domestic waste is not sorted and will cause environmental pollution (AC1)
	Dumping of domestic waste without sorting will result in waste of resources (AC2)
Ascription of Responsibility (AR)	I am responsible for the destruction of environmental quality caused by the sorting of domestic waste (AR1)
	I am responsible for the waste of resources caused by the sorting of domestic waste (AR2)
Personal Norm (PN)	I think it is necessary to sort domestic waste (PN1)
	I think I have a responsibility to sort domestic waste (PN2)
Attitude towards the Behavior (AT)	I think that the sorting of domestic waste is correct and worth promoting (AT1)
	Sorting of domestic waste is conducive to improving the governance level of human settlements (AT2)
Social Norm (SN)	My family thinks that domestic waste should be sorted (SN1)
	My neighbor thinks that domestic waste should be sorted (SN2)
Perceived Behavioral Control (PBC)	I have the time and ability to do simple sorting of domestic waste (PBC1)
	I have experience of simple sorting of domestic waste (PBC2)
Behavioral Intention (BI)	I am willing to sort and recycle domestic waste (BI1)
	I plan to sort domestic waste in the future (BI2)

**Table 2 ijerph-17-04887-t002:** Variable factor analysis and reliability and validity test results.

Construct	Item	Loadings	AVE	Cronbach’s Alpha	CR
Awareness of Consequences (AC)	AC1	0.770	0.7105	0.825	0.8298
	AC2	0.910			
Ascription of Responsibility (AR)	AR1	0.780	0.5663	0.764	0.7228
	AR2	0.724			
Personal Norm (PN)	PN1	0.798	0.6392	0.786	0.7799
	PN2	0.801			
Attitude towards Behavior (AT)	ATT1	0.634	0.5258	0.675	0.6862
	ATT2	0.806			
Social Norm (SN)	SN1	0.801	0.6456	0.796	0.7846
	SN2	0.806			
Perceived Behavioral Control (PBC)	PBC1	0.833	0.6914	0..818	0.8175
	PBC2	0.830			
Behavioral Intention (BI)	BI1	0.868	0.7363	0.851	0.8481
	BI2	0.848			

AVE: Average Variance Extracted; CR: Composite Reliability.

**Table 3 ijerph-17-04887-t003:** Differentiated validity test results.

Construct	AC	SN	AR	PBC	AT	PN	BI
AC	0.7105						
SN	0.3457	0.6456					
AR	0.5170	0.5610	0.5663				
PBC	0.2070	0.5975	0.3352	0.6914			
AT	0.4543	0.4396	0.3807	0.2621	0.5258		
PN	0.3058	0.6194	0.5344	0.3697	0.3136	0.6456	
BI	0.2938	0.5098	0.4199	0.3931	0.3982	0.6368	0.7363

Note: The data on the diagonal in the table comprise the mean variance extractions (AVE) of each latent variable, while the remaining data under the diagonal comprise the squares of the correlation coefficients between the latent variables.

**Table 4 ijerph-17-04887-t004:** Structural equation modeling (SEM) estimation results.

Hypothesis	Standardized Regression Weights	C.R.	*p*-Value	Result
H1: AT → BI	0.240	3.043	0.002	Supported
H2: SN → BI	−0.055	−0.354	0.724	Not Supported
H3: PBC → BI	0.185	2.023	0.043	Supported
H4: SN → AT	0.407	4.754	***	Supported
H5: SN → PBC	0.773	11.557	***	Supported
H6: AC → AR	0.425	6.039	***	Supported
H7: AR → PN	0.323	3.317	***	Supported
H8: PN → BI	0.594	5.284	***	Supported
H9: AC → AT	0.435	5.128	***	Supported
H10: AC → SN	0.588	8.807	***	Supported
H11: SN → AR	0.499	6.804	***	Supported
H12: SN → PN	0.544	5.420	***	Supported

**Note:** *** *p* < 0.001.

**Table 5 ijerph-17-04887-t005:** Direct effect, indirect effect, and total effect test result.

Path	Direct Effect	Indirect Effect	Total Effect
H1: AT → BI	0.240 **	0.000	0.240 **
H2: SN → BI	−0.055	0.660 ***	0.605 ***
H3: PBC → BI	0.185 *	0.000	0.185 *
H4: SN → AT	0.407 ***	0.000	0.407 ***
H5: SN → PBC	0.773 ***	0.000	0.773 ***
H6: AC → AR	0.425 ***	0.284 ***	0.719 ***
H7: AR → PN	0.323 **	0.000	0.323 **
H8: PN → BI	0.594 *	0.000	0.594 ***
H9: AC → AT	0.435 ***	0.239 ***	0.674 ***
H10: AC → SN	0.588 ***	0.000	0.588 ***
H11: SN → AR	0.499 ***	0.000	0.499 ***
H12: SN → PN	0.544 ***	0.161 **	0.706 ***

Note: *** *p* < 0.01, ** *p* < 0.05, * *p* < 0.1.

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
