# Peer review of "Investigating Rural Domestic Waste Sorting Intentions Based on an Integrative Framework of Planned Behavior Theory and Normative Activation Models: Evidence from Guanzhong Basin, China"

_ijerph, 2020, doi:10.3390/ijerph17134887_

Round 1

Reviewer 1 Report

The paper is interesting. The reviewer sees that the paper has some meanings for waste management.

The followings are comments from the reviewer.

  • On Lines 107–115, 140–144, 167–173, H1–H12 are needed just after Hypothesis 1­Hypothesis 10, respectively, for corresponding to Table 4.

For examples, they are like Hypothesis 1 (H1), Hypothesis 2 (H2), and so on.

  • For Figure 1, the abbreviations such as AC, AR, PN, and so on are needed corresponding to Table 2.

For examples, they are like Awareness of Consequences (AC), Ascription of Responsibility (AR), and so on.  The same is for Figure 3.

Why total questionnaires were just 350 people?

Author Response

Comment 1: On Lines 107–115, 140–144, 167–173, H1–H12 are needed just after Hypothesis 1¬Hypothesis 10, respectively, for corresponding to Table 4.

Response: Thanks for the reviewer’s kind suggestion. We have made correction according to the reviewer’s comments. The revised details can be found in Lines 111–119, 144–148, 171–177, page 3-4.

Comment 2: For Figure 1, the abbreviations such as AC, AR, PN, and so on are needed corresponding to Table 2. For examples, they are like Awareness of Consequences (AC), Ascription of Responsibility (AR), and so on. The same is for Figure 3.

Response: Thanks for the reviewer’s suggestion. According to his/her advices, we have made correction on Figure 2 and Figure 3, which can be find in page 5 and 9.

Comment 3: Why total questionnaires were just 350 people?

Response: Thanks for the reviewer’s question. We are very sorry for the sample size. Please allow me to explain that since the place we investigated in July is just the season for peach harvest, the purchasers from all over the country go to the countryside to collect peach in those days, and the farmers are eager to sell the peach to the purchasers in a limited time. It takes some time to do the questionnaire survey, some farmers may not be willing to accept the questionnaire survey, so we only collect 350 questionnaires at present. We hope to select more regions and samples in future research. We hope reviewer can understand.

Reviewer 2 Report

This paper entitled “Investigating rural domestic waste sorting intentions based on an integrative framework of planned behavior theory and normative activation models: Evidence from Guanzhong Basin, China.” It is based on planned behaviour theory (TPB) and normative activation theory (NAM), analyzes the domestic waste sorting intentions of rural residents in Guanzhong, China. Based on the micro-data of rural households in China, structural equation modelling was constructed to empirically analyze the influencing factors of farmers’ domestic waste sorting intention. I found the authors' research study interesting and appreciate the level of detail provided in the “hypothesis proposed section”. However, there are certain areas that require improvement in order to render the manuscript suitable for publication. Moreover, major revisions of language and style are suggested as shown as follow:

  1. It was suggested to describe the research contribution in the abstract paragraph.
  2. In the introduction section, the authors must explain the trends and social issues of rural domestic waste disposal in the world, not just China.
  3. What is the representativeness of the research scope of Heyang County, Guanzhong Plain, China?
  4. The authors should provide a more favourable explanation for the number of samples.
  5. Results and discussion lack coherence. It is suggested to use research data for analysis.
  6. In the conclusion section, the authors should make policy recommendations based on the research results.
  7. H2:SN→BI (Line 290) the result was not Supported, what does it stand for?
  8. Some spelling errors make it difficult to read. This article recommends professional English editing.

Author Response

Comment 1: It was suggested to describe the research contribution in the abstract paragraph.

Response: Thanks for the reviewer’s kind suggestion. We added the research contribution in revised manuscript and the detailed revision can be found in Line 11-27, Page 1.

Comment 2: In the introduction section, the authors must explain the trends and social issues of rural domestic waste disposal in the world, not just China.

Response: Thanks for the reviewer’s suggestion. We added this point into our revised manuscript and the details can be found in Line 32-47, Page 1-2.

Comment 3: What is the representativeness of the research scope of Heyang County, Guanzhong Plain, China?

Response: Thanks for the reviewer’s question. Heyang County is located in the northeast of Guanzhong Plain, Shaanxi Province, China. It is a model county in China in terms of rural development and improvement of the ecological environment. The selection of this research area can provide some practical reference and experience for other rural areas, so that the government can better popularize the rural household waste sorting policy. The details can be found in Line 200-206, Page 5.

Comment 4: The authors should provide a more favourable explanation for the number of samples.

Response: Thanks for the reviewer’s kind advice. At the time of the survey, farmers were busy harvesting peaches and selling them to purchasers in the countryside. Some farmers may not be willing to take the time to accept the questionnaire survey, so we only collected 350 questionnaires at present. If there is a chance in the future, we will choose a broader research scope and sample size for the study. I have added explanations in the discussion section. Hope the reviewer understand.

Comment 5: Results and discussion lack coherence. It is suggested to use research data for analysis.

Response: Thanks for the reviewer’s suggestion. We have revised this part in revised manuscript and the details can be found in Line 266-291, 299-326, 330-349, Page 7-10.

Comment 6: In the conclusion section, the authors should make policy recommendations based on the research results.

Response: Thanks for the reviewer’s kind advice. We added the policy recommendations into our revised manuscript and the details can be found in Line 359-379, Page 10-11.

Comment 7: H2:SN→BI (Line 290) the result was not Supported, what does it stand for?

Response: Thanks for the reviewer’s question. The empirical results show that hypothesis 2 has not been verified, indicating that the direct impact of farmers' subjective norms on their domestic waste sorting intentions is not significant, which is consistent with the research results of Bamberg and Mser. However, through empirical testing, subjective norms have an indirect impact on farmers' domestic waste sorting intentions through attitudes, perceived behavior control, and personal norms. This shows that the ineffectiveness of subjective norms on the direct impact of domestic waste sorting will not necessarily prevent it from indirectly enhancing the role of farmers' domestic waste sorting intentions through other influencing factors. On the contrary, subjective norms increase farmers' intentions to sort domestic waste by forming a positive attitude and a strong sense of personal responsibility. Moreover, combined with effect analysis, we know that the total impact of subjective norms on the intentions to sort domestic waste is the largest. Bamberg and Mser, Park and Ha also confirmed in previous research that subjective norms play a key role in domestic waste sorting. The details can be found in the results and discussion section of our revised manuscript.

Comment 8: Some spelling errors make it difficult to read. This article recommends professional English editing.

Response: Thanks for the reviewer’s careful check. We feel sorry for the inconvenience brought to the reviewer. We have sought professional English editors to correct this article based on the reviewer's suggestions.

Round 2

Reviewer 2 Report

In this revised version, the authors made helpful revisions. However, there are certain areas that require improvement in order to render the manuscript suitable for publication. Moreover, major revisions of language and style are suggested as shown as follow:

  1. Some classic arguments are suggested, such as:
    “Collection of domestic waste. Review of occupational health problems and their possible causes. Science of the total environment170(1-2), 1-19. ”
    “Psychosocial Risk, Work-Related Stress, and Job Satisfaction among Domestic Waste Collectors in the Ho Municipality of Ghana: A Phenomenological Study.  J. Environ. Res. Public Health2020, 17, 2903.”
  2. Some spelling errors make it difficult to read. This article recommends professional English editing.

Author Response

Comment 1: Some classic arguments are suggested, such as:

“Collection of domestic waste. Review of occupational health problems and their possible causes. Science of the total environment, 170(1-2), 1-19.”

“Psychosocial Risk, Work-Related Stress, and Job Satisfaction among Domestic Waste Collectors in the Ho Municipality of Ghana: A Phenomenological Study.  J. Environ. Res. Public Health2020, 17, 2903.”

Response: Thanks for the reviewer’s suggestion. According to his/her advices, we have added three references related to this article, including the two references recommended by the reviewer and another one with classical arguments, and the details can be found in References 5, 19 and 53.

Comment 2: Some spelling errors make it difficult to read. This article recommends professional English editing.

Response: Thanks for the reviewer’s careful check. We feel sorry for the inconvenience brought to the reviewer. We have sought professional English editors to correct this article again based on the reviewer's suggestions.